# Effects of Memantine in Patients with Traumatic Brain Injury: A Systematic Review

**Sungeen Khan** [1] , **Ayesha S. Ali** [2] , **Bryar Kadir** [2] , **Zubair Ahmed** [1,3,*] **and Valentina Di Pietro** [1,3]

1 Neuroscience and Ophthalmology, Institute of Inflammation and Ageing, University of Birmingham, Birmingham B15 2TT, UK; sungeen.khan@nhs.net (S.K.); v.dipietro@bham.ac.uk (V.D.P.)
2 Cancer Research UK Clinical Trials Unit (CRCTU), Institute of Cancer and Genomic Sciences, University of Birmingham, Birmingham B15 2TT, UK; a.s.ali@bham.ac.uk (A.S.A.); b.kadir@bham.ac.uk (B.K.)
3 Surgical Reconstruction and Microbiology Research Centre, National Institute for Health Research, Queen Elizabeth Hospital, Birmingham B15 2TH, UK
* Correspondence: z.ahmed.1@bham.ac.uk

**Abstract:** Traumatic brain injury (TBI) affects millions of people around the world and amongst other effects, causes cognitive decline, neurodegenerative disease and increased risk of seizures and sensory disturbances. Excitotoxicity and apoptosis occur after TBI and are mediated through the N-methyl-D-aspartate (NMDA)-type glutamate receptor. Memantine is effective in blocking excessive activity of NMDA-type glutamate receptors and reduces the progression of dementia and may have benefits after TBI. Here, we performed a systematic review of the literature to evaluate whether memantine is effective in improving outcomes, including cognitive function in patients with TBI. Our search yielded only 4 randomized control trials (RCTs) that compared the effects of memantine to placebos, standard treatment protocols or piracetam. A single RCT reported that serum neuron-specific enolase (NSE) levels were significantly reduced ($p = 0.009$) in the memantine compared to the control group, and this coincided with reported significant day-to-day improvements in Glasgow Coma Scale (GCS) for patients receiving memantine. The remaining RCTs investigated the effects of memantine on cognitive function using 26 standardized tests for assessing cognition function. One RCT reported significant improvements in cognitive function across all domains whilst the other two RCTs, reported that patients in the memantine group underperformed in all cognitive outcome measures. This review shows that despite laboratory and clinical evidence reporting reduced serum NSE and improved GCS, supporting the existence of the neuroprotective properties, there is a lack of reported evidence from RCTs to suggest that memantine directly leads to cognitive improvements in TBI patients.

**Keywords:** memantine; cognitive function; neuroprotection; traumatic brain injury; head injury

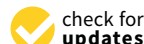



## 1. Introduction

In 2016, more than 27 million new cases of traumatic brain injuries (TBIs) were diagnosed globally, while approximately an equal number of patients were suffering from the sequelae of older TBIs [1]. Several national studies have established that there is a significant financial burden related to TBIs due to associated healthcare costs and loss of earnings/productivity [2,3]. Financial losses for a proportion of TBI patients continue after their initial hospitalization period, as they suffer from the sequelae of traumatic brain injury [4]. The sequelae of TBI can include cognitive decline, behavioural changes, neurodegenerative disease processes, motor deficits, somnolence, hormonal dysfunction, increased risk of seizures and sensory disturbances [5–8]. Cognitive impairments after TBI can include disturbances of attention, memory and executive function, resulting in reduced global cognition, naming, incidental memory, immediate memory, learning and delayed recall [9–13].

In this systematic review, we focus on whether an Alzheimer's disease (AD) medication, memantine hydrochloride (memantine), produces benefits in TBI patients. TBI can be divided into three major categories: (i) closed head; (ii) penetrating; and (iii) explosive blast TBI [14,15]. Closed head TBI typically occurs after blunt impact incurred through motor vehicle accidents, falls and sporting activities and leads to immediate damage of the brain vasculature and neurons. Penetrating TBI results from foreign body penetration of the skull and brain parenchyma causing focal damage, intracranial haemorrhage, edema and ischemia [14,15]. Explosive blast TBI however, is prevalent in war-related casualties and compromises brain tissues due to the rapid pressure shock waves generated from explosions leading to widespread diffuse damage such as neuronal death, axonal injury, compromised blood–brain barrier and edema [14,15].

Excitotoxicity and apoptosis are two mechanisms of neuronal cell death that occur in TBI, with the N-methyl-D-aspartate (NMDA)-type glutamate receptors implicated in both mechanisms [16–22]. With moderate hyperactivity of glutamate receptors, there is an excessive influx of calcium ($Ca^{2+}$) which leads to apoptosis (programmed cell death) [16,17]. Whereas, in excitotoxicity, there is a massive release of glutamate resulting in the loss of $Mg^{2+}$ within the glutamate receptor's ion channel [15]. Without the regulating effect of $Mg^{2+}$, there is an influx of calcium and sodium, which causes the neuronal cells to depolarize, swell and lyse (necrosis) [16,17]. With necrosis, there is a release of cellular contents that leads to neighbouring neuronal dysfunction or neuronal cell death by excitotoxicity. Neuronal dysfunction occurs secondary to ischemia caused by the increased energy demands needed to maintain ion gradients [16,17]. Similarly, activation of NMDA receptors by glutamate promotes the production of reactive oxygen species (ROS) and nitric oxide (NO) which further exacerbate secondary cell injury [17–19]. Memantine blocks excessive activation of NMDA-type glutamate receptors (NMDAR) since it is an uncompetitive open channel blocker which binds in the region of $Mg^{2+}$, but has a higher affinity than $Mg^{2+}$ [17–22]. In normal physiological states, the NMDA-type glutamate receptor is not open long enough to allow memantine to accumulate in its active site. Being an uncompetitive antagonist, memantine's efficacy increases as the concentration of glutamate increases [17–22].

Memantine was not only neuroprotective in animal models of cerebral and spinal cord ischemia but also in models of TBI [23–31]. Studies have also shown that blocking NMDAR function with antagonists such as amantadine, improve cognitive outcomes after mild TBI [32,33]. Hence, randomized control trials (RCTs) have been carried out to assess whether memantine has similar benefits in patients with TBI. The aim of the present study was to systematically review the data from RCTs and evaluate the efficacy of memantine in improving cognitive function and thus offering neuroprotection in patients with TBI.

## 2. Materials and Methods

### 2.1. Literature Search

The Institutional Review Board (IRB) approval was not required because this analysis was conducted on anonymized published data from the literature. The Preferred Reporting Items for Systematic reviews and Meta-analysis (PRISMA) statement [34] for systematic reviews was adhered to in creating this systematic review. When searching for appropriate RCTs to include for this systematic review Pubmed, Cinahl, Embase (OVID), PsychInfo (OVID), Cochrane (OVID) and Medline (R) and In-process and other non-indexed citations (OVID) are the databases that were searched. The terms 'memantine', 'traumatic brain injury', 'head injury', 'head injuries' and 'brain injury' are the search terms applied to databases. Searched terms were entered as text words and the search was applied to all fields, rather than just the title. This was done to ensure that appropriate studies did not go undiscovered. Boolean operators were used in structuring the search as follows: (("Memantine" [TEXT]) AND ("Traumatic Brain Injury" [TEXT] OR "Head Injury" [TEXT] OR "Head Injuries" [TEXT] OR "Brain Injury" [TEXT])). Results from searches were copied into a Microsoft Excel (Microsoft Corporation, Redmond, WA, USA) spreadsheet.

*2.2. Selection Criteria*

Two authors (S.K., and V.D.P.) assessed the title and abstract independently to select the eligible studies. The selection criteria were developed based on the following:

(1) (patients) target population must solely be adult patients with TBI;
(2) (intervention) memantine as a monotherapy in the therapeutic arm;
(3) (comparator interventions) placebo or standard treatment;
(4) (outcomes) cognitive tests, GCS and serum neuron-specific enolase;
(5) (methods-study design) RCTs;
(6) (time or duration) no specified follow-up time.

*2.3. Data Collection*

Two researchers independently extracted the following clinical data from selected studies: Basic characteristics of studies (author, year, blinding, time frame from injury, the severity of the head injury, study design, treatment regimen and outcome measures, number of participants and their gender, mechanism of injury where provided and adverse events). Relevant information from each RCT was extracted and placed into these tables. Articles were then re-read, to extract additional information regarding findings and limitations.

Data also included whether comparisons were made to a control group, placebo group or a group receiving another pharmaceutical agent. Description of the individual outcomes measures were read to reveal score ranges and how a favourable result for each outcome was defined. Comparisons were then made to determine which group performed relatively better and what trends appeared. When provided, the *p*-value was used to determine which trends were significant. When not provided the *p*-value was calculated.

*2.4. Risk of Bias*

Risk of bias was assessed using the revised Cochrane risk-of-bias tool for randomized trials (RoB 2) [35]. Each study was assessed for risk across the 5 domains by two reviewers independently (S.K. and Z.A.). Disagreements were settled through discussion. Domains assessed the potential risk of bias from the randomization process, the effect of assignment and adherence to the intervention, missing outcome data, measurement of outcome and selection of reported results. Using the algorithms following each domain, it was then determined if a study was either low risk, some risk or high risk.

*2.5. Statisitical Analysis*

When available *p*-values along with mean values of the raw data and standard deviations were used to determine when changes in outcome became significant. When *p*-values were not provided, the given mean and standard deviations from the therapeutic and control arm were processed with an online GraphPad tool [36] to calculate the Welch's *t*-test value and the two-tailed *p*-value for each outcome measure. Welch's *t*-test and the two-tailed *p*-value were used because the variances between both arms of the RCT were different. Results formed are based on reported significant changes demonstrated across included RCTs, unless specified otherwise. Upon reviewing data available by our clinical statisticians, A.A. and B.K., it was deemed that the homogeneity of outcomes was insufficient to produce a meta-analysis.

**3. Results**

*3.1. Study Selection*

The search yielded 627 results across the 6 databases. Of the 627 studies, 28 were RCTs and of these 28 RCTs, 22 were excluded on the basis that the target population was not exclusively TBI patients. A further RCT was excluded as its intervention was not a monotherapy of memantine. Of the five articles, one was excluded as it was a conference abstract of an RCT and the full paper was unavailable. Figure 1 gives an overview of the search.

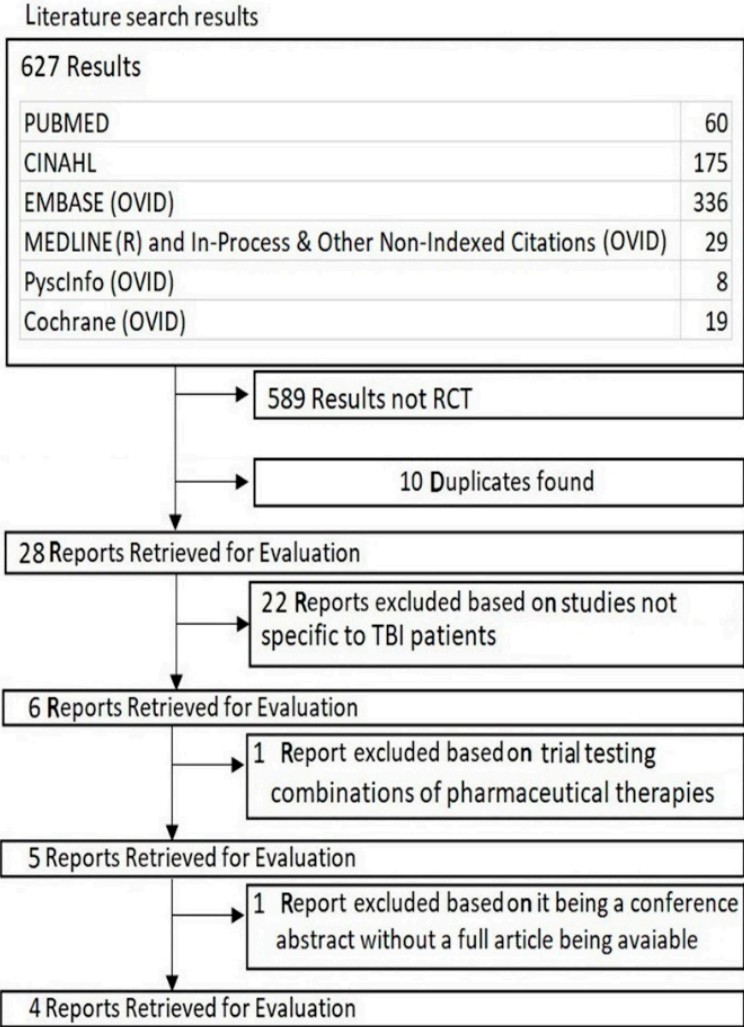

**Figure 1.** Preferred Reporting Items for Systematic reviews and Meta-analysis (PRISMA) flow chart detailing the specific of the systematic review.

*3.2. Study Characteristics*

Of the included studies, all were RCTs with the intervention in their therapeutic arms being memantine offered as a monotherapy, and with a target population that is exclusively TBI patients. Outcome measures assessed by the included RCTs were aimed at assessing either neuroprotection or cognitive functions. The onset of treatment from TBI event, the severity of TBI, sample size, study design, drug regimen and outcome measures differed across the RCTs. Both Tables 1 and 2 give an overview of the study characteristics.

**Table 1.** Characteristics of the included studies.

| Study | Study Year | Blinding | Time from TBI Event | Severity of TBI | Study Design | Drug Regimen | Outcome Measures |
|---|---|---|---|---|---|---|---|
| Mokhtari et al. [37] | 2017 | • Investigator<br>• Care provider | Within 24 h | Moderate TBI | RCT—parallel group | Memantine 30 mg twice daily for 7 days | Serum Neurons Specific Enolase<br>GCS |
| Litvinenko et al. [38] | 2010 | No Mention of Blinding | >6 Months<br>Average is 2.5 Years<br>95% ranged from 1.5 to 3.4 years | 73% Severe TBI<br>22% Moderate TBI<br>5% Mild TBI | RCT—parallel group<br>Control received piracetam | Memantine titrated up to 10 mg twice daily over 4 weeks. Piracetam 2.4 g per day | Mattis Dementia Scale<br>MMSE<br>S-test<br>FAB<br>HAM-D<br>Clock drawing |
| Rupright & Johnstone [39] | 2013 | • Participant<br>• Investigator<br>• Outcome Assessor | More than 1 year | Mild or Moderate TBI | RCT—cluster group with cross over design<br>Control received placebo | Titrated up to 20 mg. Duration is 12 weeks, followed by 4 weeks washout then cross over | Verbal Memory (HVLT-R TRLS, HVLT-R DRS)<br>Visual Memory (BVMT-R TRS, BVMT-R DRS)<br>Processing speed (TMT-A)<br>Attention (TMT-B)<br>Memory and Processing speed (SDMT-W, SDMT-O) |
| Hammond [40] | 2017 | • Participant<br>• Care Provider<br>• Investigator<br>• Outcome Assessor | Within 48 h | Severe TBI | RCT—parallel group<br>Control received placebo | (Days 1 to 3 and Days 21 to 168) 10 mg twice daily<br>(Days 3 to 21) 20 mg twice daily | Verbal Memory CVLT-II LD FR, CVLT-II T1-5 FR)<br>Visual Memory (BVMT-R DR, BVMT-R Learning)<br>Attention (TMT-B)<br>Attention, cognitive flexibility and processing speed (S1)<br>Impulse Control (BRIEF Inhibit)<br>Anger (TBI-QOL Anger) |

Traumatic brain injury (TBI), randomized control trial (RCT), Glasgow Coma Scale (GCS), mini mental state exam(MMSE), rrontal assessment battery(FAB), Hamilton depression rating score(HAM-D), Hopkin verbal learning test-revised (HVLT-R), brief visuospatial memory test—revised (BVMT-R), TMT (trail making test (TMT), symbol digit modality test—written/ oral (SDMT-W/O), total recall (learning) score (TR(L)S), delayed recall (score) (DR(S)), California verbal learning test (CVLT-II), long delay (LD), free recall (FR), trails 1 to 5 (T1–5), stroop interference (SI), behaviour rating inventory of executive function (BRIEF), and quality of life (QOL).

**Table 2.** Study year, demographics, mechanism of injury, blinding and adverse events.

| Study | Demographics and Mechanism of Injury (MOI) | | Adverse Events | |
|---|---|---|---|---|
| | Therapeutic Arm | Control Arm | Therapeutic Arm | Control Arm |
| Mokhtari et al. [37] | 22 People<br>● 21 (95.45%) Males<br>● 1 (4.55%) Female<br><br>MOI<br>3 fall1<br>8 motor vehicle accident | 19 People<br>● 18 (94.74%) Males<br>● 1 (5.26%) Female<br><br>MOI<br>3 fall<br>16 motor vehicle accident | No mention of adverse events | No mention of adverse events |
| Litvinenko et al. [38] | ● 20 People<br>● 14 (70.00%) Males<br>● 6 (30.00%) Females<br><br>No mention of MOI | ● 21 People<br>● 15 (71.43%) Males<br>● 6 (28.57%) Females<br><br>No mention of MOI | 1 patient developed anxiety | 2 patients developed anxiety |
| Rupright and Johnstone [39] | Cross over design<br>● 11 People<br>● 9 (81.82%) Males<br>● 2 (18.18%) Females<br><br>No mention of MOI | | ○ Upper respiratory tract infection (27%)<br>○ Headache (18%)<br>○ Nausea & Vomiting (18%)<br>○ Haematuria (9%) | ○ Upper respiratory tract infection (27%)<br>○ Headache (18%)<br>○ Stomach cramps (9%)<br>○ Anxiety (9%)<br>○ Urinal urgency (9%)<br>○ Sore Throat (9%) |
| Hammond [40] | At Initial enrolment<br>● 5 People<br>● 5 (100%) Males<br>● 0 Females<br>After withdrawals removed<br>- 3 People<br>- 3 Males<br>No mention of MOI | At Initial enrolment<br>● 6 People<br>● 4 (66.67%) Males<br>● 2 (33.33%) Females<br>After withdrawals removed<br>- 4 people<br>Either<br>4 Males, 0 Females or<br>3 Males, 1 Females or<br>2 Males, 2 Females<br>No mention of MOI | ○ Pneumonia (60%)<br>○ Dehydration (60%)<br>○ Neurostorming (60%)<br>○ Vomiting (40%)<br>○ Headache (40%)<br>○ Seizures (20%)<br>○ Haematuria (9.09%) | ○ Pneumonia (60%)<br>○ Neurostorming (33%)<br>○ Headache (33%)<br>○ Fall (33%)<br>○ Anxiety (9%)<br>○ Depression (9%) |

### 3.3. Results of Individual Studies

It is noted that the all the RCTs [37–40] are underpowered and therefore, where *p*-values are provided and seem to show significant findings, there remains a possibility that findings discovered are the result of chance. Equally, significant findings may go undiscovered for the same reason. RCTs included in this study are being treated as exploratory RCTs, and their results ought to be interpreted with caution.

The RCT by Mokhtari et al. [37] investigated the neuroprotective effects of memantine in TBI patients who had presented to the hospital, where this trial was taking place, within 24 h of their injury. The RCT demonstrated that from day 0 to day 7 serum neuron-specific enolase, an enzyme released following neuron damage, was significantly reduced ($p = 0.009$) in the memantine group compared to the control group. This change in neuron-specific enolase was correlated to a significant day to day improvement in the Glasgow Coma Scale scores of TBI patients ($p = 0.02$).

The RCT by Litvinenko et al. [38] utilized the Mattis dementia scale to assess cognitive function in both the memantine group and the control group. Domains of the Mattis dementia scale includes attention, initiation/perseveration, praxis, conceptualization and memory. The control group received piracetam instead of memantine. It was reported that the memantine group saw significant ($p < 0.05$) and sustained improvements across all domains of the Mattis dementia scale except praxis. The earliest significant improvement occurred in the domains of conceptualization and memory, and by the 24th week of the trial, the greatest improvements were in these two domains ($p < 0.01$). In the control group, sustained significant improvements were not reported in any domain. In the memantine group, subjective improvement of symptoms was reported by 75% of participants, compared to 57% of participants in the control groups. Participants in the memantine group demonstrated significant improvements in their performances on the mini-mental state exam, frontal assessment battery, s-test and clock drawing. Such an improvement did not occur in the control group.

The RCT by Rupright and Johnstone [39] reported no significant improvement in any of the outcome measures. The outcome measures aimed to assess verbal memory, visual memory, speed of processing and attention. When comparing mean baseline scores to those recorded at week 12 of the trial, the placebo group outperformed the memantine group in the following outcomes; the revised Hopkin verbal learning tests for total recall learning and delayed recall, the revised brief visuospatial memory tests for both total recall and delayed recall, and both the written and oral symbol digit modality tests. Trail making tests A and B showed better mean scores in the memantine group compared to the placebo group, although this improvement was not significant. Trial making tests A and B assess the speed of processing and attention, respectively.

The RCT by Hammond [40] also reported no significant improvement in outcomes. Outcomes aimed to measure verbal memory, visuospatial memory, attention, cognitive flexibility, processing speed, impulse control and anger. This RCT had only 3 participants in the therapeutic arm. The trial provided no baseline measurements, only providing a single result, at the end of the 24-week trial, under each outcome for each arm of the trial.

### 3.4. Synthesis of Results

In 3 of the 4 RCTs [38–40], all of which assess the effects of memantine on the cognitive function of TBI patients, a total of 70 participants were enrolled. A total of 34 individuals are in the therapeutic arms with the following split; 26 (76.47%) males and 8 (23.53%) females. Similarly, there are 36 participants in the control arms of these RCTs, with at least 26 (72.22%) males, 8 females (22.22%) and 2 individuals for whom there is a lack of data to determine their gender. All the participants are above the age of 18, and approximately 53% of participants had sustained a severe TBI in their lifetime. The time from TBI to treatment varied from 24 h to 3.4 years and most patients in the trial had severe or moderate TBI [37–40].

In the therapeutic arms of these RCTs [38–40], the mean scores belonging to 20 of the 34 participants, all enrolled in the RCT by Litvinenko et al. [38], indicated sustained improvements in memory, attention, initiation/perseveration and conceptualization. The combined mean scores of the remaining 14 participants showed no significant improvement in any of the standardized tests used to assess aspects of cognition. When these 14 participants were compared to their respective control groups, which received placebos in lieu of memantine, it was noted that the control group outperformed the memantine groups in the domains of verbal memory, visual memory, cognitive flexibility, processing speed, impulse control and anger. It is only in the domain of attention that the 14 participants had better scores than the placebo groups. Although the difference was not reported as significant, it was reproduced across two separate RCTs [39,40].

Across the RCTs [38–40] assessing cognitive function, participants in the therapeutic arms received doses of memantine that were titrated to a daily total of 20 mg. Duration of treatment regimens ranged from 1 month to 5.5 months.

When assessing neuroprotective properties of memantine in TBI patients 22 individuals were placed in the therapeutic arm, while 19 people were in the control arm. Each arm of this RCT [37] had a single female participant. All participants had sustained a moderate TBI up to 24 h before being enrolled in this trial. There was a significant reduction of neuron-specific enolase (NSE) in the group receiving memantine, which may indicate a neuroprotective effect in these patients. This also coincided with a significant day to day improvement in GCS score. Between all 4 RCTs assessed in this systematic review, 111 TBI patients were included and 56 were treated with memantine. Up to 42 of these 52 patients demonstrated significant favourable outcomes. However, not one of these individuals were followed up beyond the time-frame of the RCT. Therefore, we do not know if the improvements lead to sustained benefits and a significant reduction in the frequency/severity of the sequelae of TBI.

Where RCTs reported adverse events, groups receiving memantine were reported to have exclusively had the following adverse events: nausea, vomiting, hematuria and dehydration. Several adverse events were common between both the therapeutic and control arms. These include: respiratory tract infections, neurostorming, headaches, insomnia, depression and anxiety. There is no significant difference in the frequency of those adverse events common to both the therapeutic and control arms.

### 3.5. Risk of Bias within Studies

Using the Cochrane RoB2 cribsheet [35], the four RCTs were assessed across 5 domains for the potential risk of introducing bias (Figure 2). All papers were deemed low risk. However, when using the RoB2 tool it was noted that across the 4 RCTs there was no information regarding missing outcome data nor any information on the randomization process beyond statements within trials mentioning that the participants were randomly allocated.

### 3.6. Risk of Bias across Studies

When provided, demographic data only pertained to participants in the initial enrolment, and thereby was not specific to those participants who partook in the complete trial. The RCT by Litvinenko et al. [38] did not divulge information on funding, while the RCT by Mokhtari et al. [37] made no mention of adverse events. The RCT by Rupright and Johnstone [39] was partly funded by a pharmaceutical company (Forest Laboratories) that produces and sells memantine.

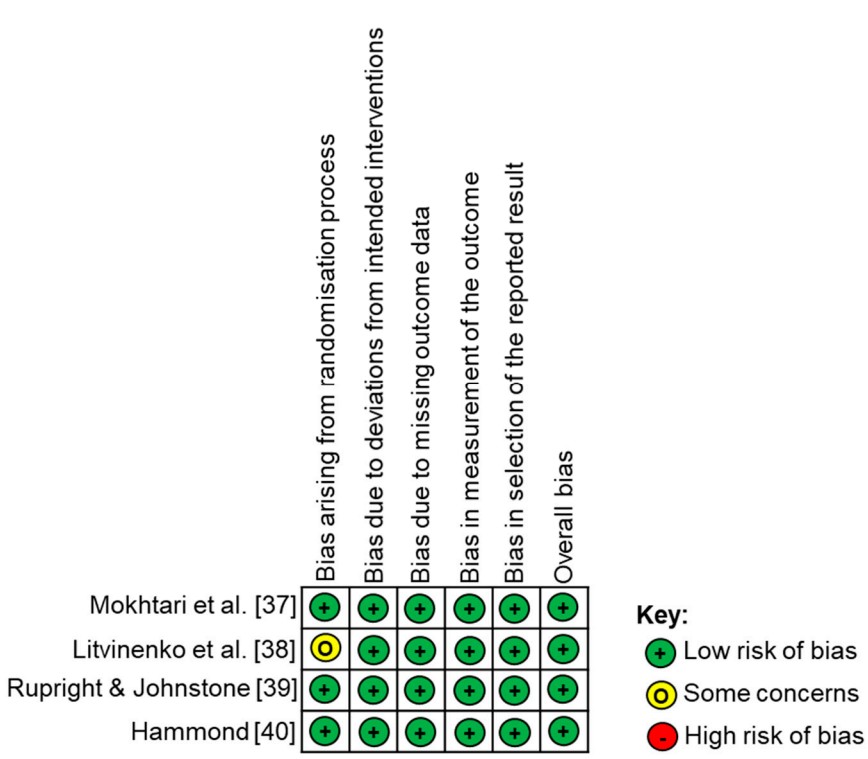

**Figure 2.** Risk of bias assessment according to the Cochrane Collaborations tool (RoB 2.0) for randomized controlled trials. Risk of bias was judged to be low.

## 4. Discussion

In this systematic review, we identified four RCTs that met our inclusion/exclusion criteria related to the use of memantine in TBI and its potential benefits on outcomes, including cognitive functions. Due to the heterogeneity within studies related to parameters including the timing of TBI to treatment and the inclusion of all forms of TBI, meta-analysis was deemed not possible and hence the four RCTs were qualitatively analysed. Our results demonstrate that in TBI, one study reported reduced serum NSE levels by day 7 and marked improvements in their GCS scores on day 3 of the study. In addition, only one study demonstrated significant improvements in cognitive outcomes across 26 standardized tests for cognitive performance, whilst two studies demonstrated that patients in the memantine group underperformed in all cognitive tests.

Across the RCTs, there were 28 outcome measures, which assessed the severity of TBI, the extent of neuronal damage, memory, cognitive flexibility, information processing, attention, conceptualization, initiation, perseverance, praxis, impulse control, depression and anger. A single RCT by Mokhtari et al. [37] presented the neuroprotective effects of memantine in TBI patients with a significant reduction of NSE and significant day to day improvement in the Glasgow Coma Scale (GCS). NSE is a commonly used as a biomarker of TBI since NSE is abundant in neuronal tissues, and structural damage of these cells cause NSE leakage into the extracellular space and into the bloodstream [41–43]. Elevated NSE levels indicate the degree of brain cell damage and correlate with unfavourable outcomes and clinical complications in neuro-intensive care units [44–50]. However, NSE levels are not 100% specific since extracranial tissues can also contribute to total serum levels if the patient suffers from severe multi-trauma or even haemolysis [51,52]. Although NSE levels correlate with mild cognitive impairment in conditions such as diabetic retinopathy and post-operative cognitive dysfunction after cardiac surgery [53,54], no relationship to NSE levels and cognitive decline have been reported in TBI [55].

The severity of TBI is defined by the duration of loss of consciousness (LOC), altered mental state (i.e., confusion) or post-traumatic amnesia (PTA) and graded according to

the GCS [56]. GCS is a 3- to 15-point scale used to assess the level of consciousness and neurological functioning and is scored on motor, verbal and eye-opening responses. In a recent study, moderate TBI patients with an initial GCS score of 9–10 exhibited greater cognitive dysfunction, compared to those with GCS scores of 11–12 [57]. Cognitive outcomes from TBI not only depend on duration of LOC and PTA but also on the degree of diffuse axonal injury, as well as evidence of brain stem dysfunction at the time of injury and the presence and size of focal hemispheric injury [58]. Since memantine increased GCS scores and higher GCS scores relate to improved LOC, an indirect effect of memantine on cognitive dysfunction may be surmised. Hence, increasing initial GCS score may reduce cognitive decline in TBI patients. Despite the fact that the study by Mokhtari et al. [37] was underpowered, the *p*-value for the reduction in serum NSE in the therapeutic arm was low ($p = 0.009$), which is promising. This study did not follow up participants after their trial period which was only 7 days, so it is difficult to establish whether NSE levels would remain lower in the treated group and whether positive effects on increasing GCS would be sustained. Furthermore, no placebo was used in the control group and so a large multicentre trial, using placebo in the comparative arm would be beneficial to confirm these findings.

All assessments demonstrating significant improvements in cognitive function were from a single RCT by Litvinenko et al. [38], the source of funding for which was not declared. Two other RCTs were carried out in America looking into cognitive improvements in TBI patients from memantine, both these studies used placebos in the control group and employed double/triple blinding. One of these RCTs, was funded by a pharmaceutical company which produced and sold memantine [39]. However, both American RCTs [39,40] did not show significant improvements in cognitive function in the memantine groups. Although the American RCTs had a superior study design, they also had less than half the sample size than in the RCT by Litvinenko et al. [38]. Clearly, this is an area where further well-controlled, suitably powered studies are required to clear up these discrepancies.

The exact dose of memantine varied slightly across the studies assessing cognitive function [38–40]. For the most part, all three RCTs used a total daily dose of 20 mg of memantine in their therapeutic arm, with 2 of the RCTs titrating to 10 mg twice daily as the dose and frequency of choice. One study used 20 mg twice daily for 19 of the 168 days of the treatment course [40]. Across the RCTs, there was no mention of improvements in cognitive function associated with as higher dose of memantine. The available raw data did not measure for the effect of dose of memantine/placebo effects on cognitive function.

The duration of therapy varied from 7 to 168 days across the 4 RCTs. RCTs employing a longer course of memantine therapy did not report significant improvement in cognitive function, and across most outcome measures, cognitive function scores were lower in assessments taken closer to the end of the drug course. Duration from injury also ranged from 48 h to 20 years amongst RCTs assessing cognitive function, with no clear benefit demonstrated when treatment with memantine was started earlier following TBI. Although in the RCT where TBI patients were treated within 48 h from the onset of TBI, only severe TBI patients were included. It is entirely possible that within this patient population there was little scope for improvements to occur.

The severity of TBI requirements across included RCT differed vastly, with the first RCT including only moderate TBI patients, the second RCT including all three severities, the third RCT excluding severe TBI patients and the final RCT selecting for severe TBI patients only. Clearly, different severities of TBI would affect the severity of cognitive decline and also the potential to recover. Closed (non-penetrating) versus open (penetrating) head injuries will also necessitate different interventions and treatments as well as different time points for intervention and hence a mixture of these patients in RCTs may confound eventual outcomes. In addition, the time from TBI event to treatment will also significantly affect the potential benefits of treatment. For example, memantine might be more effective in the acute stages after TBI in inhibiting current flow through the NMDA receptor, as a result of excessive activation by e.g., glutamate, whilst in chronic stages, memantine may

not have any effects since glutamate levels may be lower. Memantine may contribute to cognitive improvements in TBI by decreasing the synaptic 'noise' resulting from excessive NMDA receptor activation [59], inhibition of β-amyloid mediated toxicity [60–63] and readjustment of the balance between inhibition and excitation on neuronal networks in the CNS [64]. Moreover, 2 of the studies used some cognitive tests that were common to both studies [39,40] whilst the study reporting improvements in cognitive function used different cognitive tests [38]. This presents problems in comparing the apparent recovery as each test measures different types of cognitive functions. It is possible that memantine positively affected performance in the cognitive tests used by Litvinenko et al. [38] but had no effect on those used by Rupright and Johnstone [39] and Hamond [40]. Future trials assessing cognitive improvement in TBI patients would need to consider all of these important points as well as benefit from using standardized and sensitive tools for assessing cognitive outcomes after TBI.

Groups receiving memantine were reported to have exclusively had the following adverse events across RCTs reporting these events; nausea, vomiting, hematuria and dehydration. Since the target population is the same across included RCTs, several adverse events were common between both the therapeutic and control arms. These include: respiratory tract infections, neurostorming, headaches, insomnia, depression and anxiety. There is no significant difference in the frequency of those adverse events common to both the therapeutic and control arms of reviewed RCTs. This is a potential indicator that memantine has not inferred protection to TBI patients in the various therapeutic arms, although it is noted that the total number of participants enrolled in the trials were too few to draw concrete conclusions.

*Limitations*

The leading limitation of this article is that there was an inadequate number of RCTs (only 4 studies) and all RCTs included small sample sizes, rendering them underpowered. None of the articles reported alpha values, presumably for the reason that their sample sizes were too small. For this reason, a meta-analysis was not possible. Other limitations included variable time frames from TBI to treatment (48 h to 20+ years), differing severities of TBI (mild, moderate or severe) were often grouped together, demographic data when provided did not relate to the participants from whom results were generated and RCTs assessing cognitive function used different outcome measures that prevented meta-analysis from being carried out. In the future, investigators should pay more attention to performing high-quality, adequately-powered RCTs to test the effectiveness of memantine in TBI outcomes.

## 5. Conclusions

In part, the results from our systematic review support the existence of memantine's neuroprotective properties in TBI patients. However, there is insufficient evidence of cognitive improvements occurring as a direct result of this intervention in the target population. Based on the mechanism of action of memantine, it is recommended that future trials should avoid using memantine in TBI patient with long-standing sequelae of their head injury, as this intervention is unlikely to reverse established neuronal damage. Further trials are needed in patients with recent TBIs with the aim of assessing if memantine as an intervention infers long term benefits that result in a significant difference in the quality of life between those in the therapeutic arm and the control arm.

**Author Contributions:** Conceptualization, S.K., Z.A. and V.D.P.; methodology, S.K., A.S.A. and B.K.; formal analysis, S.K., A.S.A., B.K. and V.D.P.; data curation, S.K., A.S.A., B.K. and V.D.P.; writing—original draft preparation, S.K. and V.D.P.; writing—review and editing, S.K., A.S.A., B.K., Z.A. and V.D.P.; supervision, Z.A. and V.D.P. All authors have read and agreed to the published version of the manuscript.

**Funding:** This research received no external funding.

**Institutional Review Board Statement:** Ethical review and approval were waived for this study as per advice from the NHS Health Research Authority (UK) decision tool, since it is a systematic review of published literature.

**Informed Consent Statement:** Patient consent was waived because no patients or members of the public were involved in the design, conduct of this study, or reporting of this research.

**Data Availability Statement:** All data generated as part of this study are included in the article.

**Conflicts of Interest:** The authors declare no conflict of interest.

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
