# Peer review of "Effects of Memantine in Patients with Traumatic Brain Injury: A Systematic Review"

_traumacare, doi:10.3390/traumacare1010001_

Round 1

Reviewer 1 Report

The work by Khan et al represents a nice overview of the potential use of memantine in TBI as a therapeutic intervention.

the authors have compiled different RCTs and assessed the beneficial effects of the use of memantine in TBI.

the work is of high value as it demonstrates that memantine would at least won't provide the same efficacy as observed in AD. however, such a conclusion should be dealt with caution.

the are far more studies for AD compared to TBI, the heterogeneity of TBI  of closed head vs open head with the different severities would necessitate different interventions and would recommend different time points for interventions. these should be discussed in the meta-analysis.

the Authors should also discuss the value of selecting Enolase as a biomarker rather than other biomarkers (UCHL1 and GFAP  tau and NFL etc) and would be of different conclusion if these biomarkers were used instead of enolase levels.

Finally, the authors have discussed the study that evaluated the lowering of GCS due to memantine intervention; this is quite interesting as GCS is based on neurological assessment and can be correlated to cognitive performance. 

again, the authors should have more synthesis on these results in their discussion. the conclusion stating that "there is a lack of reported evidence from RCTs to suggest that memantine leads to cognitive improvements in TBI patients" should be rephrased to include the word "direct" memantine leads to direct cognitive improvements...In the future, GCS, enolase for other outcomes may be correlated to cognitive outcomes.

Author Response

Reviewer 1

  1. Comment: The work is of high value as it demonstrates that memantine would at least won't provide the same efficacy as observed in AD. however, such a conclusion should be dealt with caution.

 Answer: We agree and we have been cautious with our conclusions regarding the use of memantine for TBI patients. See Line 30 as an example.

  1. Comment: There are far more studies for AD compared to TBI, the heterogeneity of TBI of closed head vs open head with the different severities would necessitate different interventions and would recommend different time points for interventions. These should be discussed in the meta-analysis.

Answer: We have now discussed the heterogeneity in TBI patients and severities in the Discussion, Lines 350-363.

  1. Comment: The Authors should also discuss the value of selecting Enolase as a biomarker rather than other biomarkers (UCHL1 and GFAP, tau and NFL etc) and would be of different conclusion if these biomarkers were used instead of enolase levels.

Answer: We have now added text to explain why NSE levels were discussed as a biomarker than others in the Discussion, Lines 296-306.

  1. Comment: Finally, the authors have discussed the study that evaluated the lowering of GCS due to memantine intervention; this is quite interesting as GCS is based on neurological assessment and can be correlated to cognitive performance. Again, the authors should have more synthesis on these results in their discussion.

Answer: We have text in the discussion, Lines 307-322 to explain why lowering GCS by memantine might be useful. 

  1. Comment: The conclusion stating that "there is a lack of reported evidence from RCTs to suggest that memantine leads to cognitive improvements in TBI patients" should be rephrased to include the word "direct" memantine leads to direct cognitive improvements...In the future, GCS, enolase for other outcomes may be correlated to cognitive outcomes.

Answer: We have amended the sentence as recommended by the reviewer to read: This review shows that despite laboratory and clinical evidence reporting reduced serum NSE and improved GCS, supporting the existence of the neuroprotective properties, there is a lack of reported evidence from RCTs to suggest that memantine directly leads to cognitive improvements in TBI patients.

Reviewer 2 Report

There is little data on effective treatment for cognitive function after TBI, so it is interesting that the authors focus on a drug not often used. This appears also in the review that very little has been done regarding TBI and memantine.

The authors aim is to study cognitive effect of memantine. In the Introduction, there is more emphasis on dementia and long-term effect after TBI and less on cognition. The authors need to give the work a clearer framework and not only cognitive effects after memantine. They need to address why memantine is of interest for treatment after TBI, and then not only in relation to dementia. What cognitive functions can be impaired after TBI, which functions are more sensitive and what can have long lasting consequences for patients? And how can that be related to treatment with memantine?

Since the studies include a range from mild to severe TBI and different evaluation methods used, it will not be possible to make an overall assessment, which the authors also report in the discussion. I appreciate that they report adverse effects they found it, which is important for a hardly used drug after TBI and risk of bias and how this was assessed.

The treatment-studies included here are both from the acute stage and after a long time and this should be discussed in terms of treatment and why use memantine in the acute and long term stage. It is also desirable to address in the discussion possible mechanisms of action that memantine may have in the acute phase as in the chronic phase and relation to cognition and treatment.

Why is the enolase article included, as it is not about cognition?

Author Response

Reviewer 2

  1. Comment: The authors aim is to study cognitive effect of memantine. In the Introduction, there is more emphasis on dementia and long-term effect after TBI and less on cognition. The authors need to give the work a clearer framework and not only cognitive effects after memantine. They need to address why memantine is of interest for treatment after TBI, and then not only in relation to dementia. What cognitive functions can be impaired after TBI, which functions are more sensitive and what can have long lasting consequences for patients? And how can that be related to treatment with memantine?

Answer: The Introduction has been significantly revised to make this study relevant to TBI. The background to the use of memantine and the cognitive functions that are affected after TBI are now introduced.  See Lines 42-54; Lines 73-75; Lines 88-90.

  1. Comment: Since the studies include a range from mild to severe TBI and different evaluation methods used, it will not be possible to make an overall assessment, which the authors also report in the discussion. I appreciate that they report adverse effects they found it, which is important for a hardly used drug after TBI and risk of bias and how this was assessed.

Answer: More details are given for the risk of bias and a new Figure 2 is added that shows bias assessments by 2 reviewers over the 5 area as in the Cochrane RoB tool 2.0. Also, please see a substantially revised discussion regarding different severities of TBI, open vs closed head injuries etc. Lines 350-363.    

  1. Comment: The treatment-studies included here are both from the acute stage and after a long time and this should be discussed in terms of treatment and why use memantine in the acute and long-term stage. It is also desirable to address in the discussion possible mechanisms of action that memantine may have in the acute phase as in the chronic phase and relation to cognition and treatment.

Answer: We have now added a new paragraph to discuss the different treatment studies used and acute versus chronic stage treatment as well as mechanisms of actions of memantine in TBI. This is now found in Lines 351-370.

  1. Comment: Why is the enolase article included, as it is not about cognition?

Answer: We have included the article about NSE due to the fact that memantine reduced NSE levels and at the same time improved GCS scores. Since GCS scores have a neuronal element and the higher the GCS score the less cognitive decline there is in TBI, we have included the article to demonstrate that memantine may have indirect effects by increasing the initial GCS and therefore potentially offering a protective role against cognitive decline. This is now discussed in Lines 296-323

Round 2

Reviewer 2 Report

The authors have responded to all comments and I have no other comments to the manuscript.